# Association of personal and professional factors, resilience and quality of life of registered nurses in a university medical city in the Kingdom of Saudi Arabia

**Abdualrahman Saeed Alshehry**[ORCID]*

College of Nursing, King Saud University, Riyadh, Saudi Arabia

* aalshehry12345@gmail.com, abdalshehri@ksu.edu.sa

**Data Availability Statement:** All relevant data are within the manuscript and its Supporting Information file.

## Abstract

This study assessed the resilience of nurses in Saudi Arabia during the corona virus 2019 (COVID-19) pandemic and examined its influence on their quality of life (QOL). A sample of 356 nurses was surveyed in this quantitative, cross-sectional study using the Resilience Scale for Nurses and the World Health Organization Quality of Life (WHOQOL-BREF) from October 2020 to March 2021. The nurses reported the highest resilience score on "situational pattern", while the lowest score was on "relational pattern." The nurses had good perceptions on their overall QOL and health and rated their "social relationship" as having the highest quality, while their "environmental" domain as having the least quality. Gender, marital status, provision of direct nursing care to COVID-19 patients, "philosophical pattern", "situational pattern" and "dispositional pattern" had multivariate impacts on the QOL dimensions. The study concluded that being resilient can positively impact the nurses' QOL during stressful situations, such as the COVID-19 pandemic.

## 1. Introduction

The coronavirus disease 2019 (COVID-19) pandemic is a significant health concern around the world [1]. While COVID-19 vaccination programs were continuously being rolled out in different parts of the globe, the emergence of various virus variants presented new challenges [2]. Since the start of the pandemic, countries worldwide, including Saudi Arabia, had implemented immediate responses to mitigate the spread of the virus and its different variants [1]. However, the influx of COVID-19 patients in hospitals necessitated more healthcare professionals and more personal protective equipment [PPE] than usual.

Nurses play a significant role in COVID-19 infection prevention, infection control, and management [3]. However, the COVID-19 pandemic brought increased workload among nurses, exposed them to a higher risk of getting infected, and threatened their overall health and well-being. Adversity and crisis created more significant work stress, burnout, and fatigue among nurses, adversely affecting their overall health and quality of life (QOL) [4, 5]. As

**Funding:** This research was funded by the Deanship of Scientific Research at King Saud University through the Research Center in the College of Nursing.

**Competing interests:** The authors have declared that no competing interests exist.

working during a pandemic is a very stressful situation among nurses, their QOL may be challenged, and negatively affecting the quality of patient care they render [5].

The World Health Organization (WHO) defines QOL as the *"individual perceptions of people about their place in life in the light of the cultural context in which they live, and in relation to goals, expectations, models and their interests"* [6]. QOL includes a combination of physical health, emotional, mental status, autonomy, social relations, personal beliefs, and environmental characteristics [6]. One factor that could influence nurses' QOL during stressful situations is resilience. Resilience is the person's ability to handle positively and cope with stressful situations despite adversities and crises [7]. According to the Nursing Model of Resilience, resilience is one's *"ability to transform disaster into a growth experience and move forward"* [8]. This model defines the characteristics of resilience into four patterns, namely the dispositional, relational, situational, and philosophical patterns that contribute independently or as a group to create a personal web of support for an individual during stressful situations [8]. Smith and colleagues [9] described resilience as a significant protecting factor by increasing hope during adversity, creating balance in life, and fostering control over stressful situations. During the COVID-19 pandemic, nurses were continuously impacted in negative ways. During that time, nurses commonly experienced psychological burdens, such as stress and anxiety [10]. There are also times that nurses were forced to socially isolate themselves due to the risk of infecting their family and friends. Also, work-related challenges, such as the inadequate supply of PPE, increased workload, longer working hours, and unsafe workplace, brought nurses to feel distressed and burn out [11]. Thus, nurses need to be resilient during stressful times like the COVID-19 pandemic to cope with these adversities and maintain favorable well-being effectively. An individual who exhibits a higher resilience level shows an increased ability to adapt and thrive under adversity [12, 13]. This could be explained as a positive adaptation process to adversity involving dynamic relations between self, other people, and its environment.

Aside from resilience, research evidence shows that other factors including personal and professional characteristics of nurses affect their wellbeing before and during the COVID-19 pandemic. For instance, Abraham and D'silva [14] reported that designation, educational qualification, social support, burnout and job satisfaction were significantly associated with QOL among 100 nurses (50 staff nurses and 50 nurse educators) in Mangalore, India. Likewise, social support was found as one of the significant predictors of QOL together with sense of coherence and spending more time at work in a total sample of 1,040 Singaporean nurses [15]. In a quantitative, cross-sectional study participated by 52 nursing faculty in rural Appalachia, West Virginia, United States of America (USA) during the pandemic, resilience was found the strongest factor to predict each of the QOL subscales along with other factors such as: experience (years of teaching) and level of teaching (at graduate or both graduate and undergraduate level) [16]. Another quantitative, cross-sectional study revealed a link between stress, self-efficacy, resilience, and QOL among 308 Spanish critical care nurses during the pandemic but did not report significant relationship with the following sociodemographic characteristics such as: age, gender, employment relationship, care for patients with COVID, number of COVID patients, workload during the pandemic, and number of working hours per week [17].

Despite empirical data that recognizes the vital role of nurses' resiliency during adversity and crisis, such data have failed to address the resilience and QOL of nurses during this pandemic. Furthermore, to the researcher's knowledge, no study had yet been conducted examining the QOL of nurses in Saudi Arabia during this pandemic and examine how their resilience and personal and professional characteristics influence their QOL. As nurses are vital human resources for the healthcare system during this ongoing pandemic, their resiliency and QOL are crucial in providing high excellent and safe patient care and in controlling the outbreak

[5]. Understanding the relationship of the personal and professional characteristics, resilience and QOL of nurses during the COVID-19 crisis can help design interventions or training programs that could ensure the wellness and health of nurses.

## 1.1 Aims

This study assessed the resilience and QOL of nurses in Saudi Arabia during the COVID-19 pandemic. It also examined how their resilience as well as their personal and professional characteristics influences their QOL amidst this public health crisis. Specifically, this study sought to:

1. Determine the relationships between the personal and professional characteristics and QOL dimensions.

2. Examine the relationship between resilience dimensions and QOL dimensions.

# 2. Materials and methods

## 2.1 Design, sample and setting

This is a quantitative and cross-sectional study, which was performed at King Saud University Medical City (KSUMC). The hospital is one of the largest healthcare facilities in the country that caters to the health care needs of COVID-19 patients. The G*Power version 3.1.9.7 software for the A-priori sample size computation was used to achieve a statistical test of multiple regression analysis having thirteen predictor variables for the QOL of nurses, including nine personal and professional characteristics and four dimensions of resilience. A minimum computed sample size of 189 was sufficient to yield a medium size effect of 0.15, at a margin of error of 5%, and confidence level (CI) of 95%. The computed sample size was doubled (n = 378) as target sample size to ensure adequate data from the nurses. Data from the actual sample of 356 nurses who were selected using convenience sampling were included in this study. The nurses included in this study met the following criteria: (1) employed in the KSUMC during the conduct of the study, (2) with at least six months of experience (i.e., completed the probationary period at KSUMC) as a clinical nurse, (3) provides direct nursing care, and (4) consented to participate in the study.

## 2.2 Data collection

Data were collected from October 2020 to March 2021 using a survey questionnaire. Due to the ongoing pandemic restrictions at the time of administering the questionnaires, the researcher coordinated with the hospital and arranged a data collection schedule. Recruitment was done online by sending the nurses invitation to participate in their social media and email. Following the hospital protocols strictly, the researcher approached those who signified to participate in the study to obtain informed consent and hand over the questionnaire. The nurses were instructed to put the answered questionnaire in the boxes put up in each nursing station in the hospital, which were emptied weekly. The survey collected data on the following variables:

**Personal and professional data.** The data collected in these variables included age, gender, marital status, nationality, religion, education, work experience, involvement in providing care to COVID-19 patients, and attendance to educational activities on COVID-19.

**Resilience.** This variable was measured using the Resilience Scale for Nurses [18]. The scale contains 19 items that assessed the resilience of clinical nurses. The scale has four factors, namely "philosophical pattern" (six items), "relational pattern" (four items), "situational

pattern" (three items), and "dispositional pattern" (six items). The scale was responded using a 5-point scale from 1 (almost never) to 5 (almost always). Scores were obtained by calculating the mean; higher mean scores signified a greater degree of resilience. The tool's validity and reliability were established in a previous study [18], with computed Cronbach's alpha of 0.938 for the full scale. For its subscale, the alpha values ranged from 0.808 to 0.922 [18]. Park et al. [18] performed the exploratory factor analysis (EFA) and confirmatory factor analysis (CFA) and established the four-factor solution of the scale.

**QOL.**   The nurses' QOL was assessed by the "World Health Organization QOL-BREF" (WHOQOL-BREF). The 26-item, 5-point Likert scale measures four dimensions, namely "physical health" (seven items), "psychological health" (six items), "social relationships" (three items), and "environment domain" (eight items). Various studies have employed this scale to measure variations in QOL across different cultures. Items 1 and 2 of the WHOQOL-BREF gauged the respondents' overall QOL and health perceptions, respectively. The items were scored from 1 to 5. Raw domain scores were transformed to scores ranging from 4–20 based on the guidelines and then summed to obtain the domain scores. The domain scores were then transformed linearly to a 0–100 scale, in which high score signified high QOL. Different studies had provided evidence of the strong validity and reliability of WHOQOL-BREF [6].

In the current study, to establish the face validity of the instruments on resilience and QOL, a pilot study was conducted among 22 nurses and were excluded in the main study. The face validity evaluated the appearance of the instruments in terms of consistency of style, formatting, readability, and respondents' understanding of the terms used in each instrument. All nurses involved in the pilot study responded that the questions and terms used in the instruments were clear and easy to read. The respondents noted that they finished answering the questionnaire in an average time of 12 minutes, ranging from 10 to 14 minutes. For the reliability of each instrument and subscales, Cronbach's alpha was calculated. The subscales of the Resilience Scale for Nurses had the following Cronbach's alpha values ranging from 0.787 to 0.935 and the Cronbach's alpha of the overall scale was 0.968. For the WHOQOL-BREF subscales, the Cronbach's alpha values ranged from 0.857 to 0.929 with an overall scale Cronbach's alpha of 0.947.

## 2.3 Ethical considerations

An Institutional Review Board (IRB) in the College of Medicine at KSUMC reviewed and granted approval to conduct the study (Reference No.: 20/0693/IRB). This research study followed the ethical principles in conducting survey studies, including the principle of justice, informed consent (written and verbal), and respect for human dignity. The principle of justice included explaining the risk (no known harm mentally, emotionally, physically or in any other aspect) and benefits (no renumeration provided for participation) of the study and respecting the respondents' culture. The respondents were also informed that their participation is kept confidential, and their responses are completely anonymous that no one would know including their nurse managers or KSUMC as their employer. After the respondents were given complete and honest information about the study and allowed to ask questions, a written informed consent was obtained from all respondents involved in the study using the informed consent form provided by the IRB. The respondents were also informed that participation is voluntary and that they can withdraw at any phase of the study without negative consequences.

## 2.4 Statistical analysis

Using the Statistical Package for the Social Sciences (SPSS) version 22.0, descriptive analyses (i.e., frequency [f], percentage [%], mean [M], range and standard deviation [SD]) were carried

out to analyze the personal and professional variables and resilience and QOL. The researcher conducted a multivariate multiple regression to inspect the multivariate effects of the nurses' personal and professional characteristics and resilience (predictor variables) on their perceptions of the four dimensions of QOL. Multiple regression analyses were then performed on each QOL dimension to further assess the specific effect of the predictor variables. Before the conduct of the regression analyses, was all personal and professional characteristics as predictor variables were dummy coded. The personal and professional variables and the four dimensions of resilience were entered as predictor variables in each regression model. Before performing the regression analyses, the assumptions for conducting them were examined first. There was no issue with multicollinearity as the Variance Inflammation Factor (VIF) values were below 4.00 and the tolerance values were above 0.20. The P-P plots showed normal distribution of the residuals of the regression (dependent variables). P values less than 0.05 will be considered significant.

## 3. Results

### 3.1. Personal and professional characteristics of nurses

Among the 356 nurses (Response rate = 94.18%) surveyed in this study, 324 (91.0%) were female, and 32 (9.0%) were males, with an average age of 36.58 years (SD = 7.54). The majority of the nurses were married (60.1%), Filipinos (80.1%), Christians (88.8%), and had a Bachelor of Science (BSN) degree (81.7%). Most of the respondents were involved in providing direct nursing care to COVID-19 patients (69.1%) and had attended educational activities on COVID-19 since the start of the pandemic (94.4%). The average year of experience of the nurses was 12.26 years (SD = 6.47) (see Table 1).

### 3.2 Results of the descriptive analyses on nurses' resilience

The results of the descriptive analyses on the Resilience Scale for Nurses are summarized in Table 2. The overall mean score of the respondents on the scale was 4.14 (SD = 0.42). Among the four dimensions of the scale, the nurses achieved the highest mean score on "situational pattern" (M = 4.23, SD = 0.52), followed by "dispositional pattern" (M = 4.14, SD = 0.52), and "philosophical pattern" (M = 4.12, SD = 0.50). The nurses rated the dimension "relational pattern" (M = 4.11, SD = 0.52) as the poorest dimension of their resilience.

### 3.3 Results of the descriptive analyses on nurses' QOL

The mean score of the nurses on the overall perceptions of their QOL and health amidst the COVID-19 pandemic was 4.23 (SD = 0.52) and 4.08 (SD = 0.58), respectively. The nurses rated their "social relationship" dimension (M = 72.66, SD = 17.47) as having the highest quality, followed by their "psychological health" (M = 71.06, SD = 11.77), then "physical health" (M = 65.35, SD = 11.21). The "environmental" dimension (M = 65.25, SD = 12.57) was perceived to be the lowest QOL dimension of the nurses during the pandemic (see Table 3).

### 3.4 Results of the multivariate regression analysis using Wilks' Lambda test on the four dimensions of QOL

Table 4 reflects the results of the multivariate multiple regression analysis on the four dimensions of QOL. The analysis identified the following predictor variables as having multivariate effect on the four dimensions of the QOL: gender ($\lambda$ = 0.95, $p$ = .002), marital status ($\lambda$ = 0.94, $p$ = .0007), provision of direct nursing care to COVID-19 patients ($\lambda$ = 0.94, $p$ = .001),

**Table 1. Personal and professional characteristics of nurses (n = 356).**

| Variable | f | % |
|---|---|---|
| Gender | | |
| Male | 32 | 9.0 |
| Female | 324 | 91.0 |
| Marital status | | |
| Single | 142 | 39.9 |
| Married | 214 | 60.1 |
| Nationality | | |
| Saudi | 20 | 5.6 |
| Filipino | 285 | 80.1 |
| Indian | 51 | 14.3 |
| Religion | | |
| Muslim | 40 | 11.2 |
| Christian | 316 | 88.8 |
| Education | | |
| Diploma/ Associate in nursing | 65 | 18.3 |
| BSN | 291 | 81.7 |
| Involved in providing direct care to COVID-19 patients | | |
| No | 110 | 30.9 |
| Yes | 246 | 69.1 |
| Attended educational activities on COVID-19 | | |
| No | 20 | 5.6 |
| Yes | 336 | 94.4 |
| | Mean (SD) | Range |
| Age (years) | 36.58 (7.54) | 25–60 |
| Work experience (years) | 12.26 (6.47) | 0.50–35 |

Note. f = Frequency, % = Percentage, SD = Standard Deviation

"philosophical pattern" ($\lambda = 0.88$, $p = .0009$), "situational pattern" ($\lambda = 0.96$, $p = .015$), and "dispositional pattern" ($\lambda = 0.95$, $p = .001$).

## 3.5 Results of the multiple regression analyses on each dimension of QOL

The regression models for each QOL dimension shown in Table 5 were statistically significant, accounting for 27.0% ($F[14, 341] = 10.36$, $p < .001$), 39.1% ($F[14, 341] = 17.28$, $p < .001$), 27.4% ($F[14, 341] = 10.57$, $p < .001$), and 25.2% ($F[14, 341] = 9.55$, $p < .001$) of the variance in the nurses' perceptions of their "physical health", "psychological health", "social relationship" and "environmental health", respectively. The values corresponding to the percentage of the variance for each dimension of QOL are not presented in Table 5. However, the values presented in Table 5 are narratively described below.

Specifically, females compared to males had higher levels of "physical health" ($\beta = 4.08$, $p = .025$, 95% confidence interval [CI] = 0.51, 7.65), "psychological health" ($\beta = 4.85$, $p = .006$, 95% CI = 1.43, 8.27), and "social relationship" ($\beta = 9.43$, $p = .001$, 95% CI = 3.89, 14.98). Nurses who were married had better "physical health" ($\beta = 2.27$, $p = .047$, 95% CI = 0.04, 4.51), "social relationship" ($\beta = 7.30$, $p = .0009$, 95% CI = 3.83, 10.78), and "environmental health" ($\beta = 2.86$, $p = .027$, 95% CI = 0.32, 5.40) than nurses who were single. Nurses who were involved in providing direct nursing care to COVID-19 patients manifested significantly poorer

**Table 2. Results of the descriptive analyses on nurses' resilience (n = 356).**

| Variable | Mean | SD |
|---|---|---|
| Dimension1: Philosophical pattern | 4.12 | 0.50 |
| 1. I feel generally happy | 3.43 | 0.90 |
| 2. I am satisfied with my life | 3.91 | 0.83 |
| 3. I am an important person | 4.10 | 0.85 |
| 4. I have hope for the future | 4.60 | 0.59 |
| 5. I have a strong goal consciousness for life | 4.50 | 0.61 |
| 6. I am a necessary person to someone else | 4.20 | 0.74 |
| Dimension 2: Relational pattern | 4.11 | 0.52 |
| 7. I lead the conversation while considering the position of the other person | 3.79 | 0.78 |
| 8. I fully accept the advice of others | 4.03 | 0.78 |
| 9. I keep a good relationship with people around me | 4.40 | 0.66 |
| 10. There are people around me to help when I have a difficult task | 4.23 | 0.69 |
| Dimension 3: Situational pattern | 4.23 | 0.52 |
| 11. I know when I am not involved in the work or I am involved | 4.13 | 0.67 |
| 12. I have the ability to determine the priority order during execution of the job | 4.24 | 0.62 |
| 13. I have the ability to determine the work that I can do or cannot do | 4.32 | 0.61 |
| Dimension 4: Dispositional pattern | 4.14 | 0.52 |
| 14. I am a strong person and can cope with life's challenges and adversity | 4.24 | 0.67 |
| 15. I can do a new job or a difficult work | 4.10 | 0.64 |
| 16. I do not give up under any circumstances | 4.20 | 0.70 |
| 17. I am working autonomously | 4.10 | 0.69 |
| 18. Once I start doing something, I can achieve the expected goal | 4.04 | 0.70 |
| 19. I have the ability to cope with stressful work situations | 4.13 | 0.67 |
| Overall resilience | 4.14 | 0.42 |

Note. SD = Standard Deviation

"physical health" ($ß$ = -5.22, p = .0006, 95% CI = -7.54, -2.90), "psychological health" ($ß$ = -2.95, p = .009, 95% CI = -5.18, -0.73), and "environmental health" ($ß$ = -2.96, p = .028, 95% CI = -5.59, -0.33) than those who were not involved in the care process of such patients.

Regarding the influence of resilience on QOL, a point increase in the nurses' mean scores on "philosophical patterns" corresponded to 4.68 (p = .0007, 95% CI = 2.12, 7.24), 8.00 (p = .0006, 95% CI = 5.54, 10.45), 7.27 (p = .0009, 95% CI = 3.29, 11.25), and 7.71 (p = .0007, 95% CI = 4.80, 10.61) points increased in the mean scores for "physical health", "psychological health", "social relationship" and "environmental health", respectively. Similarly, an increase

**Table 3. Results of the descriptive analyses on nurses' QOL (n = 356).**

| Variables | Range | | Mean | SD |
|---|---|---|---|---|
| Physical health | 32.14 | 92.86 | 65.35 | 11.21 |
| Psychological health | 25.00 | 100.00 | 71.06 | 11.77 |
| Social relationship | 16.67 | 100.00 | 72.66 | 17.47 |
| Environmental | 21.88 | 96.88 | 65.25 | 12.57 |
| Overall perception of their QOL | 3.00 | 5.00 | 4.23 | 0.52 |
| Overall perception of their health | 2.00 | 5.00 | 4.08 | 0.58 |

Note. SD = Standard Deviation

**Table 4. Results of the multivariate regression analysis using Wilks' Lambda test on the four dimensions of QOL (n = 356).**

| Predictor variables | Value | F value | Hypothesis df | Error df | p value |
|---|---|---|---|---|---|
| Age | 1.00 | 0.36 | 4.00 | 339.00 | .840 |
| Gender | 0.95 | 4.30 | 4.00 | 339.00 | .002** |
| Marital status | 0.94 | 5.22 | 4.00 | 339.00 | .0007*** |
| Nationality | 1.00 | 0.20 | 4.00 | 339.00 | .939 |
| Religion | 0.98 | 1.61 | 4.00 | 339.00 | .172 |
| Education | 0.98 | 1.77 | 4.00 | 339.00 | .135 |
| Work experience | 0.99 | 0.64 | 4.00 | 339.00 | .638 |
| Provided care to COVID-19 patients | 0.94 | 5.07 | 4.00 | 339.00 | .001** |
| Attended educational activities on COVID-19 | 0.99 | 0.64 | 4.00 | 339.00 | .636 |
| Philosophical pattern | 0.88 | 12.03 | 4.00 | 339.00 | .0009*** |
| Relational pattern | 0.98 | 2.17 | 4.00 | 339.00 | .073 |
| Situational pattern | 0.96 | 3.13 | 4.00 | 339.00 | .015* |
| Dispositional pattern | 0.95 | 4.96 | 4.00 | 339.00 | .001** |

Note. *Significant at .05

**Significant at .01

***Significant at .001, df = Degrees of Freedom

in the nurses' mean scores on "relational pattern" was linked to increased mean scores on "psychological health" (ß = 2.61, p = .044, 95% CI = 0.07, 5.16) and "social relationship" (ß = 5.75, p = .006, 95% CI = 1.63, 9.87). Better resilience in terms of "situational pattern" predicted a better "physical health" (ß = 3.39, p = .010, 95% CI = 0.80, 5.98), whereas a better "dispositional pattern" resilience predicted a better "psychological health" (ß = 3.73, p = .005, 95% CI = 1.16, 6.31).

## 4. Discussion

This study assessed the nurses' resilience and examined how it influences their QOL amid the ongoing COVID-19 pandemic. The study reported an overall high resilience among nurses, which concurs with the high level of resilience exhibited by nurses in the same country in the study of Balay-odao et al. [10]. However, different scales were used in the two studies. In comparing the present finding with other studies conducted in other countries, the nurses in this study manifested higher levels of resilience during this pandemic than those in China [19] and the United Kingdom (UK) [20, 21]. The high levels of resilience in this study indicate that nurses could cope holistically with the stresses brought about by the COVID-19 pandemic. Despite the various challenges that the COVID-19 pandemic had brought in the nurses' personal and professional, they were able to remain positive. They showed capabilities to surpass these challenges while continuously exhibiting professional nursing competence in the clinical settings [18]. The pandemic has brought extreme stresses and anxieties to nurses, in addition to their social isolation [22]. Since the start of the pandemic, nurses are confronted with issues (i.e., risk of exposure, fear of bringing home the virus, and anxiety on dealing with infected patients) brought about by the pandemic itself that is physically, emotionally, mentally, socially, and spiritually draining [11]. The ability of the nurses to cope effectively in these difficult situations is being manifested in their highest resilience on the dimension "situational patterns." This dimension of resilience encompasses the nurses' capabilities to effectively process stressful situations and to be flexible and patient in coping with them [18]. Several factors may have contributed to the high resilience of the nurses, such as personal (i.e., self-care,

**Table 5. Results of the multiple regression analyses on each dimension of QOL (n = 356).**

| Dependent variable | Predictors | ß | SE | p value | 95% CI | |
|---|---|---|---|---|---|---|
| | | | | | Lower | Upper |
| Physical health $R^2$ (Adjusted $R^2$) = 0.298 (0.270) | Age | 0.04 | 0.13 | .741 | -0.21 | 0.30 |
| | Gender | 4.08 | 1.81 | .025* | 0.51 | 7.65 |
| | Marital status | 2.27 | 1.14 | .047* | 0.04 | 4.51 |
| | Nationality (Ref.: Filipino) | | | | | |
| | Saudi | 3.27 | 2.66 | .219 | -1.95 | 8.50 |
| | Indian | 0.37 | 2.14 | .864 | -3.84 | 4.57 |
| | Religion | 0.92 | 1.91 | .630 | -2.83 | 4.67 |
| | Education | -0.05 | 1.92 | .978 | -3.82 | 3.71 |
| | Work experience | 0.04 | 0.15 | .789 | -0.26 | 0.34 |
| | Involved in providing direct care to COVID-19 patients | -5.22 | 1.18 | .0006*** | -7.54 | -2.90 |
| | Attended educational activities on COVID-19 | -1.82 | 2.30 | .430 | -6.33 | 2.70 |
| | Philosophical pattern | 4.68 | 1.30 | .0007*** | 2.12 | 7.24 |
| | Relational pattern | 1.82 | 1.35 | .178 | -.83 | 4.48 |
| | Situational pattern | 3.39 | 1.32 | .010* | 0.80 | 5.98 |
| | Dispositional pattern | 2.27 | 1.37 | .098 | -0.42 | 4.96 |
| Psychological health $R^2$ (Adjusted $R^2$) = 0.415 (0.391) | Age | 0.14 | .124 | .268 | -0.11 | 0.38 |
| | Gender | 4.85 | 1.739 | .006** | 1.43 | 8.27 |
| | Marital status | 0.63 | 1.090 | .566 | -1.52 | 2.77 |
| | Nationality (Ref.: Filipino) | | | | | |
| | Saudi | -1.13 | 2.55 | .658 | -6.13 | 3.88 |
| | Indian | -1.51 | 2.05 | .463 | -5.54 | 2.53 |
| | Religion | -1.09 | 1.83 | .551 | -4.68 | 2.50 |
| | Education | 1.37 | 1.84 | .456 | -2.24 | 4.98 |
| | Work experience | 0.10 | 0.15 | .509 | -0.19 | 0.38 |
| | Involved in providing direct care to COVID-19 patients | -2.95 | 1.13 | .009** | -5.18 | -0.73 |
| | Attended educational activities on COVID-19 | 1.14 | 2.20 | .604 | -3.19 | 5.47 |
| | Philosophical pattern | 8.00 | 1.25 | .0006*** | 5.54 | 10.45 |
| | Relational pattern | 2.61 | 1.29 | .044* | 0.07 | 5.16 |
| | Situational pattern | 1.79 | 1.26 | .156 | -0.69 | 4.28 |
| | Dispositional pattern | 3.73 | 1.31 | .005** | 1.16 | 6.31 |
| Social relationship $R^2$ (Adjusted $R^2$) = 0.303 (0.274) | Age | 0.10 | 0.20 | .624 | -.30 | 0.49 |
| | Gender | 9.43 | 2.82 | .001** | 3.89 | 14.98 |
| | Marital status | 7.30 | 1.77 | .0009*** | 3.83 | 10.78 |
| | Nationality (Ref.: Filipino) | | | | | |
| | Saudi | 0.03 | 4.12 | .995 | -8.09 | 8.14 |
| | Indian | -2.43 | 3.32 | .464 | -8.97 | 4.10 |
| | Religion | -2.72 | 2.96 | .359 | -8.55 | 3.10 |
| | Education | -4.32 | 2.97 | .147 | -10.17 | 1.53 |
| | Work experience | 0.19 | 0.24 | .411 | -0.27 | 0.66 |
| | Involved in providing direct care to COVID-19 patients | -2.89 | 1.83 | .116 | -6.50 | 0.72 |
| | Attended educational activities on COVID-19 | -0.58 | 3.57 | .871 | -7.60 | 6.44 |
| | Philosophical pattern | 7.27 | 2.02 | .0009*** | 3.29 | 11.25 |
| | Relational pattern | 5.75 | 2.10 | .006** | 1.63 | 9.87 |
| | Situational pattern | 5.89 | 2.04 | .004** | 1.87 | 9.91 |
| | Dispositional pattern | -3.59 | 2.12 | .092 | -7.76 | 0.59 |

(*Continued*)

**Table 5.** (*Continued*)

| Dependent variable | Predictors | ß | SE | *p value* | 95% CI | |
|---|---|---|---|---|---|---|
| | | | | | Lower | Upper |
| Environmental $R^2$ (Adjusted $R^2$) = 0.282 (0.252) | Age | 0.11 | 0.15 | .468 | -0.18 | 0.39 |
| | Gender | 1.40 | 2.06 | .498 | -2.65 | 5.44 |
| | Marital status | 2.86 | 1.29 | .027* | 0.32 | 5.40 |
| | Nationality (Ref.: Filipino) | | | | | |
| | Saudi | 2.23 | 3.01 | .459 | -3.69 | 8.15 |
| | Indian | -0.71 | 2.43 | .769 | -5.48 | 4.06 |
| | Religion | -4.15 | 2.16 | .056 | -8.40 | 0.11 |
| | Education | 0.58 | 2.17 | .791 | -3.70 | 4.85 |
| | Work experience | -0.12 | 0.17 | .501 | -0.45 | 0.22 |
| | Involved in providing direct care to COVID-19 patients | -2.96 | 1.34 | .028* | -5.59 | -0.33 |
| | Attended educational activities on COVID-19 | -2.05 | 2.61 | .432 | -7.17 | 3.07 |
| | Philosophical pattern | 7.71 | 1.48 | .0007*** | 4.80 | 10.61 |
| | Relational pattern | 2.83 | 1.53 | .065 | -0.18 | 5.84 |
| | Situational pattern | 1.53 | 1.49 | .306 | -1.41 | 4.47 |
| | Dispositional pattern | 1.45 | 1.55 | .349 | -1.59 | 4.50 |

Note. *Significant at .05

**Significant at .01

***Significant at .001, $R^2$ = Coefficient of determination, ß = Beta value, SE = Standard Error, CI = Confidence Interval

mindfulness, social support) and organizational (i.e., supportive working environment, adequate resources for PPE, prompt and effective communication and information provision) factors [11]. However, such factors were not investigated in this study; thus, prompting future investigations.

Among the four dimensions of QOL, the nurses reported their social relationship as having the highest quality, while the environmental domain was perceived as the poorest domain. Social relationships as the highest domain came as a surprise considering the social restrictions and isolation that are being implemented to combat the spread of the virus. However, this finding may signify that the nurses could cope effectively with these restrictions and find meaning in the social support from their family, friends, co-workers, and supervisors in these difficult times [23]. The mean scores across the four domains of QOL indicate that the nurses had moderate to good levels of QOL based on the recommended interpretations of the WHO-QOL-BREF scores in previous studies [18, 24]. However, interpreting the WHOQOL-BREF results in this manner should be taken with caution since there is no consensus on how to interpret its finding and the WHO have not recommended such interpretations. Thus, comparing the present findings on previous QOL scores among nurses could provide a better understanding of the level of QOL of the present sample in this pandemic. For instance, the present QOL in terms of the four dimensions were slightly better than those reported among nurses in Singapore before the pandemic [15] and in Taiwan during the pandemic [25], but were poorer when compared with the QOL reported among nursing faculty members in the US during the pandemic [16]. Also, when compared to nurses in Saudi Arabia before the pandemic, the nurses in the present study had perceived their QOL more inferior in all dimensions [5], but with better QOL levels compared to healthcare workers (HCWs) across 19 countries in the Arab region, two years after the pandemic started [26]. The impact of the COVID-19 pandemic on everybody's health, well-being, and QOL are undeniable. This

pandemic had caused nurses and other HCWs to suffer by being infected with the virus [27]. Moreover, this pandemic caused nurses to experience poor mental health conditions, poor well-being, decreased compassion satisfaction and increased burnout and secondary traumatic stress [28, 29]. These factors could have direct and indirect effects on how nurses perceived their QOL amidst the COVID-19 pandemic.

Furthermore, the study's findings provide valuable evidence on the positive impact of resilience on QOL during difficult situations, such as the COVID-19 pandemic. However, this finding provided contradicting evidence on the relationship of resilience and QOL in comparison to the finding of a previous study conducted in Saudi Arabia [30]. Grande et al.'s [30] study found no association between academic resilience and QOL among nursing students. The present findings showed that during this pandemic, the nurses' personal beliefs and optimism for the future (philosophical pattern), their high regards to friendship and authentic relationships (relational pattern), their self-confidence and self-efficacy to overcome obstacles (dispositional pattern), and their capabilities in interpreting and coping stressful situations with flexibility and patience (situational pattern) play critical roles in maintaining good levels of QOL in one or more dimensions. For instance, having positive beliefs and being optimistic amidst this pandemic contributed to better physical and psychological health, social relationships, and environmental health. A previous study had argued that being optimistic and less pessimistic during the COVID-19 pandemic aid in coping with coronavirus stress and promotes lower levels of psychological problems [31]. In Cruz et al.'s [32] study, optimistic nurses displayed a better capability to effectively utilize coping strategies when faced with stressful situations, which eventually impacted their QOL. Another study reported that having effective self-efficacy was associated with lesser use of passive coping and more frequent use of active coping and consequently causes lesser mental health problems [33], which could explain how self-efficacy and self-confidence result in better psychological health. Also, acknowledging stressful situations and effectively coping with them could directly or indirectly affect one's physical health. For example, Ito and Matsushima [34] supported that negative coping and positive coping may lead to poorer and better physical health, respectively. Overall, the findings strengthen the evidence on the protective role of resilience during stressful situations, such as the COVID-19 pandemic. High levels of resilience have buffering effects on an individual's general health, experiences of anxiety, and stress during stressful life events [12]. Psychological resilience is an essential element for successful crises management among HCWs and enables them to recover more rapidly and effectively from the negative impacts of the pandemic through developing cognitive, emotional, and interpersonal skills that promote effective adaptive coping behaviors [35].

Another strong predictor of the nurses' QOL during this pandemic is their experience of providing care to COVID-19 patients. The findings indicated that nurses involved in providing direct care to COVID-19 patients were more likely to experience poorer physical, psychological, and environmental health than those without similar experiences. The experience of being the frontline HCWs during this pandemic is challenging and exhausting for nurses [36]. With the mounting numbers of infected individuals, the COVID-19 pandemic had challenged the capacity and resources of hospitals, which leave HCWs to be physically, mentally, emotionally, and spiritually exhausted. Moreover, the frequent exposure of nurses to infected patients constantly threatens their risk and increasing their fear of infecting their family and friend. Their experience of COVID-19 related illness and deaths of a family member, a friend, a colleague, or a patient also negatively impacts their overall well-being [37]. These factors may have influenced the poor QOL of nurses working directly with COVID-19 patients. Surprisingly, female nurses reported better physical and psychological health and social relationships than male nurses, contrary to previously reported gender differences on QOL in studies

conducted in Saudi Arabia [5, 38]. Having better QOL among female nurses in the present findings may be related to the ongoing transformation in the country, which aims to empower women in all aspects of society [39]. Moreover, the findings of the current study are consistent with a recent study among 322 frontline nurses in Saudi Arabia which reported the following sociodemographic characteristics such as: age, gender, marital status, years of work experience, extra working hours, and direct contact with COVID patients were significant predictors of the nurses' QOL during the pandemic [40]. However, these findings must be interpreted with caution as other confounding variables reported in previous studies in Saudi Arabia such as eating bahavior, shift duty and stress [41] and sleep quality [42], may have impacted the QOL of registered nurses.

While the study presents a critical investigation on the well-being of nurses during this pandemic, several limitations should be considered in interpreting and discussing the study's findings. The study's cross-sectional design compared to longitudinal design limited its ability to examine causal relationships between resilience and QOL; hence, the predictive relationship between the two main variables was focused. The single setting included in the study and the convenience sampling technique utilized for sample selection may have caused the inability of the findings to represent the entire nursing population in the country. This prevented generatability of the study findings to nurses working in other government hospitals and private hospitals in Saudi Arabia as well as in the Arab countries and internationally. In addition, the use of self-reported data from the nurses posed social desirability response bias. Nevertheless, the contributions of this study on the current concern of nurses' well-being during this pandemic should not be underestimated.

## 5. Conclusions

The study concludes that the nurses exhibited resilience amidst all the difficulties brought by the COVID-19 pandemic. They were also able to maintain moderate to good levels of QOL despite the ongoing pandemic. The study further concludes that being resilient can positively impact the physical and psychological health, social relationships, and environmental health of nurses during stressful situations, such as the COVID-19 pandemic. Moreover, providing direct nursing care to patients with COVID-19 infection increases nurses' risk of experiencing poor physical, psychological, and environmental health. Therefore, the findings provide critical information that healthcare facilities leaders and policymakers could use to ensure that nurses enjoy excellent levels of QOL through fostering resilience amidst the COVID-19 pandemic.

## 6. Implications and recommendations

The findings of the current study provide several implications to the nursing practice and strengthen the evidence on the critical role of resilience in ensuring high levels of QOL among nurses during stressful situations, such as the COVID-19 pandemic. For instance, the study emphasizes the importance of ensuring the excellent well-being of nurses during the COVID-19 pandemic and calls for hospital leaders and policymakers to prioritize this aspect. The significant effects of resilience to the QOL reported in the study could provide the basis for creating programs in the hospitals to build and sustain nurses' resilience during stressful situations in the country, among Arab countries and internationally.

Several measures on building and maintaining resilience among HCWs during the pandemic had been recommended based on literature review, such as providing accurate and timely information, psychosocial support and management, monitoring regularly the health status of healthcare professionals, effective management and delegation of tasks and

responsibilities, and managing work patterns and working conditions [43]. These recommendations could be considered in developing programs specific to the needs of the nurses and the clinical environment. Moreover, special attention is needed for nurses who are directly involved in providing care to COVID-19 patients as they are at higher risk of experiencing lower levels of QOL. Programs and interventions tailored to their needs while providing care to COVID-19 patients are recommended to ensure that they can cope effectively with the stresses brought about by their roles and responsibilities while providing patient care during stressful situations like the COVID-19 pandemic and in preparation for future pandemics. Future research including longitudinal studies can be implemented to identify causality between resilience and QOL, and track changes within the study sample until post-pandemic time. Lastly, to enhance the validity of the findings, the use of objective measures or triangulation with other data sources such as using qualitative design to explore experiences of nurses related to resilience and QOL during the pandemic can be conducted in future studies.

## Supporting information

**S1 File. This is the raw data.**
(XLSX)

## Author Contributions

**Conceptualization:** Abdualrahman Saeed Alshehry.

**Data curation:** Abdualrahman Saeed Alshehry.

**Formal analysis:** Abdualrahman Saeed Alshehry.

**Funding acquisition:** Abdualrahman Saeed Alshehry.

**Investigation:** Abdualrahman Saeed Alshehry.

**Methodology:** Abdualrahman Saeed Alshehry.

**Project administration:** Abdualrahman Saeed Alshehry.

**Resources:** Abdualrahman Saeed Alshehry.

**Software:** Abdualrahman Saeed Alshehry.

**Supervision:** Abdualrahman Saeed Alshehry.

**Validation:** Abdualrahman Saeed Alshehry.

**Visualization:** Abdualrahman Saeed Alshehry.

**Writing – original draft:** Abdualrahman Saeed Alshehry.

**Writing – review & editing:** Abdualrahman Saeed Alshehry.

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
