## [Decision Letter · Decision Letter 0]

16 Jun 2024

PONE-D-24-08992The influence of personal and professional factors, and resilience on the quality of life of registered nurses in the Kingdom of Saudi ArabiaPLOS ONE

Dear Dr. Alshehry,

Thank you for submitting your manuscript to PLOS ONE. After careful consideration, we feel that it has merit but does not fully meet PLOS ONE’s publication criteria as it currently stands. Therefore, we invite you to submit a revised version of the manuscript that addresses the points raised during the review process.

Please address reviewers’ comments.

We look forward to receiving your revised manuscript.

Kind regards,

Majed Sulaiman Alamri, PhD

Academic Editor

PLOS ONE

Journal Requirements:

3. In the online submission form, you indicated that [Data are available on request from the author.]. 

Reviewers' comments:

Reviewer's Responses to Questions

**Comments to the Author**

1. Is the manuscript technically sound, and do the data support the conclusions?

Reviewer #1: Partly

Reviewer #2: Yes

Reviewer #3: Yes

Reviewer #4: Yes

2. Has the statistical analysis been performed appropriately and rigorously? 

Reviewer #1: No

Reviewer #2: Yes

Reviewer #3: I Don't Know

Reviewer #4: Yes

3. Have the authors made all data underlying the findings in their manuscript fully available?

Reviewer #1: No

Reviewer #2: Yes

Reviewer #3: Yes

Reviewer #4: No

4. Is the manuscript presented in an intelligible fashion and written in standard English?

Reviewer #1: Yes

Reviewer #2: Yes

Reviewer #3: Yes

Reviewer #4: Yes

5. Review Comments to the Author

Reviewer #1: Cross-Sectional Design Limitations

The use of a cross-sectional design limits the ability to infer causality between resilience and quality of life (QOL). While it is acknowledged that longitudinal designs may be more challenging to implement, it is important to clearly state this limitation and discuss how it affects the interpretation of the results. Additionally, the authors should consider suggesting longitudinal studies in the future research section to provide stronger evidence for causality.

Convenience Sampling Bias

The use of convenience sampling raises concerns about the representativeness of the sample. This method can introduce selection bias, limiting the generalizability of the findings. The authors should acknowledge this limitation more explicitly in the manuscript and discuss its potential impact on the study's conclusions.

Self-Reported Data Concerns

Reliance on self-reported data can introduce response biases, such as social desirability bias and recall bias. The manuscript should discuss these limitations and consider suggesting the use of objective measures or triangulation with other data sources in future research to enhance the validity of the findings.

Lack of Clear Topic Sentences

Several paragraphs lack clear and concise topic sentences, making it difficult for readers to follow the logical flow of the manuscript. Each paragraph should begin with a strong topic sentence that clearly introduces the main idea. The authors should revise these sections to improve readability and coherence.

Imprecise Reporting of Statistical Results

The statistical results reported in this study are sometimes imprecise. For example, stating the exact p-values (e.g. p = 0.032) instead of "<.05" would enhance transparency. Additionally, some tables lack sufficient details regarding the statistical tests used, making it challenging to fully interpret the results. The authors should provide more comprehensive details in the tables and the text.

Ethical Considerations and Anonymity

While the manuscript mentions that written informed consent was obtained, details on measures taken to ensure the anonymity and confidentiality of participants' data are also included. This information is crucial for maintaining ethical standards, and should be explicitly stated in the manuscript.

Risks to Participants

Potential risks to the participants were not adequately discussed. A thorough explanation of any risks involved in participation along with measures taken to mitigate these risks should be included to ensure adherence to ethical standards.

Validation of Measurement Tools

The manuscript mentions the validity and reliability of the Resilience Scale for Nurses and WHOQOL-BREF but lacks specific details on how these were confirmed in the current study population. Providing detailed psychometric properties specific to this sample would enhance the credibility of the instrument used.

Translation Process

The translation process for any non-English version of the questionnaire used was not described. This is a critical omission because ensuring the validity of the translated instruments is essential. The authors should include a description of the translation and validation process.

Linking Discussion to Research Questions

There are instances in which the discussion is not clearly linked to the study's aims and hypotheses. For example, the exploration of gender differences in QOL should be more explicitly connected to the research questions posed at the beginning of the study. The authors should ensure that all discussions are tied to the initial research objectives.

Justification for Predictor Variables

The rationale for including specific predictor variables in the regression models is inadequately justified. The manuscript should provide a clear and compelling rationale for the inclusion of each predictor variable, as supported by relevant literature.

Global Significance of Findings

While the focus on the Saudi Arabian context is justified, this manuscript should better articulate the global significance of the findings. Highlighting the broader implications and potential applications of the results would strengthen the contribution of the manuscript to the field.

Detailed Justification of Hypotheses

The introduction lacks detailed justification for the specific hypotheses tested. A more thorough explanation of the theoretical underpinnings and the empirical evidence guiding the hypotheses would provide a stronger foundation for this study.

Discussion of Limitations

The Limitations section is too brief and does not adequately discuss how each limitation impacts the findings. A more detailed analysis of the study's limitations and suggestions for mitigating these issues in future research are necessary.

Specific Future Research Recommendations

Recommendations for future research are generally general and lack specificity. Providing targeted and actionable suggestions based on the identified gaps and limitations is valuable.

Assumptions of Multivariable Multiple Regression

The manuscript uses multivariable multiple regression appropriately, but lacks details on the assumptions checked before performing these analyses (e.g. multicollinearity and normality). Providing this information is crucial to assess the robustness of the statistical methods used.

Avoiding "Impact" in the Title

The use of the term "impact" in the title is misleading, as the study's design does not allow causality to be established. A more accurate term would be "association" or "relationship.

Articulating Study's Importance

While this study addresses a significant topic, its importance could be better articulated in terms of its potential influence on policy and practice. The manuscript should emphasise how the findings can inform healthcare policies and interventions in more detail.

Recommendation:

There are substantial areas that require improvement, particularly in methodology, reporting, and discussion. Therefore, we recommend that the manuscript be revised with major changes before it can be considered for publication. These revisions should address methodological weaknesses, enhance the clarity and coherence of the writing, and provide a more robust discussion and justification of the findings.

Reviewer #2: The manuscript is well written and it discussed an important topic. The introduction section outlines the important of the topic, the methods section addressed all required elements, the results section presents the findings in a readable manner

Reviewer #3: hank you for giving me the opportunity to review this manuscript. The authors made a good effort in writing this study. However, there are some comments about the quality of the research and its presentation in a suitable way for the readers.

Please justify selecting nurses with experience of at least six months.

Add more information about the sample size calculation.

The sample was selected from a single hospital; therefore, the title of the paper is supposed to reflect that instead of generalizing this to Saudi Arabia. 

Studies related to COVID-19 are out of date; however, the author may link the clinical implications of this study to more applicable lessons for any health crisis.

Reviewer #4: This study examines the resilience and quality of life (QOL) of nurses in Saudi Arabia during the COVID-19 pandemic. The topic is relevant given the significant impact the pandemic has had on healthcare workers globally. The study methodology appears sound, using validated scales and appropriate statistical analyses. The manuscript is organized and clearly written.

Abstract:

- The abstract is concise and provides a good overview of the study.

- Add “cross-sectional” to the statement “in this quantitative study”.

Methods

- “Ethical Considerations” section is too long. Some statements are redundant and not needed. For example, “The principle of justice also included the right to privacy. The respondents were not harmed mentally, emotionally, or in any other aspect”.

- Always mention the full term of any abbreviation when first appears in the text. For example, “EFA and CFA” in line 118.

- Author stated, “Before the conduct of the regression analyses, the predictor variable nationality was dummy coded.” I think all categorical/binary predictors were dummy coded.

Discussion:

- When comparing your results to prior studies, it is important to note if findings align with or differ from those reported from the same region such as: (https://pubmed.ncbi.nlm.nih.gov/35983443) who also examined quality of life and academic resilience in Saudi nursing.

- The discussion is missing the discussion of potential confounders that may affect QOL of nurses other than those addressed in the current study. For example, occupational factors such as shift work and lifestyle factors such as sleep, diet, and stress have been reported from the same country:

o https://pubmed.ncbi.nlm.nih.gov/26837403/

o https://pubmed.ncbi.nlm.nih.gov/38549756/

- Conclusion section is very long. It can be shortened.

6. PLOS authors have the option to publish the peer review history of their article (what does this mean?). If published, this will include your full peer review and any attached files.

Reviewer #1: **Yes: **Mosharop Hossian

Reviewer #2: **Yes: **Homood Awadh Alharbi

Reviewer #3: No

Reviewer #4: No

---

## [Author Response · Author response to Decision Letter 0]

12 Aug 2024

PONE-D-24-08992R1

Association of personal and professional factors, resilience and quality of life of registered nurses in a university medical city in the Kingdom of Saudi Arabia

Dr Abdualrahman Saeed Alshehry

RE: Response to Edits Requested on August 07, 2024

Dear Ronalyn M. Ramos,

Thank you very much for your editorial feedback with edits requested.

1.Can you please upload an additional copy of your revised manuscript that does not contain any tracked changes or highlighting as your main article file. This will be used in the production process if your manuscript is accepted. Please amend the file type for the file showing your changes to Revised Manuscript w/tracked changes. Please follow this link for more information: http://blogs.PLOS.org/everyone/2011/05/10/how-to-submit-your-revised-manuscript/

RESPONSE: Thank you for this feedback. May I clarify that I already resubmitted an additional copy of my revised manuscript as CLEAN copy with no tracked changes with File Name: ‘Revised Manuscript_CLEAN_PlosOne_Alshehry_V1’. I am resubmitting it again after complying your feedback/comment below – to delete ethics statement at the end of the main text.

2.Your ethics statement should only appear in the Methods section of your manuscript. If your ethics statement is written in any section besides the Methods, please delete it from any other section.

RESPONSE: Thank you for this feedback. The other ethics statement (Lines 505-508) of my revised manuscript with tracked changes has been deleted, as required.

I am looking forward to merit your kind assistance and to hear from you very soon.

Very sincerely,

Abdualrahman S. Alshehry

---

## [Decision Letter · Decision Letter 1]

28 Aug 2024

Association of personal and professional factors, resilience and quality of life of registered nurses in a university medical city in the Kingdom of Saudi Arabia

PONE-D-24-08992R1

Dear Dr. Alshehry, 

We’re pleased to inform you that your manuscript has been judged scientifically suitable for publication and will be formally accepted for publication once it meets all outstanding technical requirements.

Kind regards,

Majed Sulaiman Alamri, PhD

Academic Editor

PLOS ONE

Additional Editor Comments (optional):

Reviewers' comments:

Reviewer's Responses to Questions

**Comments to the Author**

1. If the authors have adequately addressed your comments raised in a previous round of review and you feel that this manuscript is now acceptable for publication, you may indicate that here to bypass the “Comments to the Author” section, enter your conflict of interest statement in the “Confidential to Editor” section, and submit your "Accept" recommendation.

Reviewer #2: All comments have been addressed

Reviewer #4: All comments have been addressed

2. Is the manuscript technically sound, and do the data support the conclusions?

Reviewer #2: Yes

Reviewer #4: Yes

3. Has the statistical analysis been performed appropriately and rigorously? 

Reviewer #2: Yes

Reviewer #4: Yes

4. Have the authors made all data underlying the findings in their manuscript fully available?

Reviewer #2: Yes

Reviewer #4: Yes

5. Is the manuscript presented in an intelligible fashion and written in standard English?

Reviewer #2: Yes

Reviewer #4: Yes

6. Review Comments to the Author

Reviewer #2: (No Response)

Reviewer #4: (No Response)

7. PLOS authors have the option to publish the peer review history of their article (what does this mean?). If published, this will include your full peer review and any attached files.

Reviewer #2: No

Reviewer #4: No

---

## [Editor Report · Acceptance letter]

1 Sep 2024

PONE-D-24-08992R1 

PLOS ONE

Dear Dr. Alshehry, 

I'm pleased to inform you that your manuscript has been deemed suitable for publication in PLOS ONE. Congratulations! Your manuscript is now being handed over to our production team.

Kind regards, 

on behalf of

Dr. Majed Sulaiman Alamri 

Academic Editor

PLOS ONE